# Individual and Combined Effects of Aflatoxin B1 and Sterigmatocystin on Lipid Peroxidation and Glutathione Redox System of Common Carp Liver

**DOI:** 10.3390/toxins13020109

**Published:** 2021-02-02

**Authors:** Benjamin Kövesi, Szabina Kulcsár, Zsolt Ancsin, Erika Zándoki, Márta Erdélyi, Miklós Mézes, Krisztián Balogh

**Affiliations:** 1Department of Nutrition, Szent István University, H-2100 Gödöllő, Hungary; benjamin.kovesi@gmail.com (B.K.); szabina.kulcsar@gmail.com (S.K.); Ancsin.Zsolt@szie.hu (Z.A.); Ballane.Erdelyi.Marta@szie.hu (M.E.); Balogh.Krisztian.Milan@szie.hu (K.B.); 2Mycotoxins in the Food Chain Research Group, Hungarian Academy of Sciences, Kaposvár Campus, Szent István University, H-7400 Kaposvár, Hungary; Zandoki.Erika@mkk.szie.hu

**Keywords:** sterigmatocystin, aflatoxin B1, oxidative stress, glutathione redox system, gene expression, common carp

## Abstract

The purpose of the study was to evaluate the short-term effects of aflatoxin B1 (AFB1 100 µg/kg feed) and sterigmatocystin (STC 1000 μg/kg feed) exposure individually and in combination (100 μg AFB1 + 1000 μg STC/kg feed) on the parameters of lipid peroxidation and glutathione redox system both in biochemical and gene expression levels in one-year-old common carp. Lipid peroxidation parameters were slightly affected, as significant differences were observed only in conjugated diene and triene concentrations. Reduced glutathione content decreased more markedly by STC than AFB1 or AFB1+STC, but glutathione peroxidase activity did not change. Expression of *gpx4a, gpx4b, gss,* and *gsr* genes was down-regulated due to STC compared to AFB1 or AFB1+STC, while an induction was found as effect of AFB1+STC in the case of *gpx4a,* but down-regulation for *gpx4b* as compared to AFB1. Expression of the glutathione biosynthesis regulatory gene, *gss*, was higher, but glutathione recycling enzyme encoding gene, *gsr,* was lower as an effect of AFB1+STC compared to AFB1. These results are supported by the changes in the expression of transcription factors encoding genes, *nrf2,* and *keap1.* The results revealed that individual effects of AFB1 and STC on different parameters are synergistic or antagonistic in multi-toxin treatment.

## 1. Introduction

Increasing fish production in aquaculture produces a high demand for fish feeds. However, fishmeal, an important ingredient of fish feeds, will be scarce in the future. Therefore, cereals are often used to replace at least a part of the fishmeal in fish feeds, leading to increased mycotoxin contamination [1,2].

Mycotoxins are secondary metabolites of filamentous fungi. Both aflatoxins (AF) and sterigmatocystin (STC) are primarily produced by molds belonging to the *Aspergillus* genus [3,4]. The four most important aflatoxins are the aflatoxin B1 (AFB1), the most toxic, aflatoxin B2 (AFB2), aflatoxin G1 (AFG1), and aflatoxin G2 (AFG2), which have slight structural differences [5]. According to the recent BIOMIN World Mycotoxin Survey Report [6], aflatoxin contamination was 8% in finished (complete) feeds and 21% in cereals, with an average of positive samples containing 10 µg/kg in finished feeds and 2 µg/kg in cereals in Europe in 2019. There are only limited data about STC prevalence in food and feed; however, it contaminates various crops, spices, brewery, and dairy products [7]. The chemical structure of STC is similar to that of aflatoxins, and it acts as a biogenic precursor in the biosynthetic pathway of aflatoxin B1 (AFB1) and aflatoxin G1 (AFG1) [8]. Despite the similar molecular structure of STC to AFB1, its acute toxicity is approximately ten times lower [9].

In vivo and in vitro studies have reported that STC has immunomodulatory and mutagenic effects on bacterial and mammalian cell lines [10,11,12,13,14]. The induction of chromosomal damages and sister-chromatid exchange was also reported in vivo and in vitro [15,16], which leads to cytotoxicity [17,18], inhibition of cell cycle and mitosis [19,20,21]. In the case of AFB1, hepatotoxic, carcinogenic, mutagenic, teratogenic, and immunosuppressive effects were demonstrated even in aquatic species [22,23].

The main target organ of AFB1-toxicity is the liver, while STC can affect both the kidney and the liver [24,25]. In the liver, both AFB1 and STC metabolized by cytochrome P450 3A4 to reactive electrophilic epoxides [26,27]. These highly unstable and thus reactive epoxides—exo-AFB1-8,9-epoxide and exo-STC-1,2-epoxide—in the target cells react with cellular macromolecules, e.g., nucleic acids (forming AFB1- and STC-adducts), proteins, and phospholipids, to induce various genetic, metabolic, signaling, and cell structure disruptions [28,29]. However, an increasing amount of evidence demonstrates equally dramatic or higher effects of AFB1 or STC on cell function and integrity by induction of oxidative stress [28,30]. Nevertheless, there are only limited data on the combined effect of AFB1 and STC despite being produced by the same fungal species and sharing the same biosynthetic pathway.

In response to the elevated levels of reactive oxygen species (ROS), the expression of antioxidant enzymes at gene and protein levels is demonstrated by the hierarchical model of oxidative stress [31]. The imbalance of the redox state of the cells regulates the protein expression and activity of the transcription factor nuclear factor-erythroid 2 p45-related factor 2 (Nrf2), the master regulator of the oxidative stress response [32]. The redox-sensitive Kelch-like ECH-associated protein 1 (Keap1)-Nrf2-ARE (Antioxidant Response Element) pathway drives the expression of antioxidant genes Keap1 and represses Nrf2 transcription protein under physiological conditions. As a response to an elevated ROS level, the Nrf2-binding cysteine side chains in Keap1 oxidize; therefore, the interaction between Nrf2 and Keap1 destabilizes Nrf2 releases and reaches ARE in the nuclei [33].

Previous studies with fish support that an individual AFB1 toxin induces oxidative stress. El-Barbary et al. [34] reported a decrease of reduced glutathione (GSH) content, catalase (CAT) activity, and total antioxidant capacity (TAC), as well as increased lipid peroxidation and glutathione peroxidase (GPx) activity and gene expression after a single intraperitoneal treatment of 6 mg AFB1/kg b.w. in Nile tilapia (*Oreochromis niloticus*). Hassan et al. [35] also observed an increase in malondialdehyde (MDA) levels and a decrease in superoxide dismutase (SOD), CAT and lysozyme activities in an 84-day long experiment with Nile tilapia consuming contaminated feed at a concentration of 3 mg AFB1/kg. Similar observations were made by Abdel-Daim et al. [36] in a 30-day long experiment with 2.5 mg AFB1/kg feed exposure in which a decrease in GSH content, GPx, SOD, and CAT activity, and an increase in MDA were observed at the applied concentration. In a 60-day long feeding experiment with juvenile grass carp (*Ctenopharyngodon idella*), feeding diets contaminated with 29, 59, 86, 110, and 147 μg AFB1/kg increased MDA and ROS levels, and decrease of SOD, CAT, GPx, glutathione-S-transferase (GST) and glutathione reductase (GR) activity, and GSH levels in both kidney and spleen were measured. Dose-dependent gene expression changes of antioxidant enzymes have also been observed [37]. A 21-day long feeding experiment with common carp (*Cyprinus carpio*) fed diets contaminated with 0.5, 0.7, and 1.4 mg AFB1/kg feed increased MDA content and CAT activity, while decreasing TAC was observed [38]. In our previous short-term (24 h) study with common carp [39] with different doses of AFB1 (100, 200, and 400 mg/kg feed), parameters of the initial phase of lipid peroxidation (conjugated dienes and trienes) increased while marker of the termination phase, thiobarbituric acid reactive substances (TBARS) expressed as MDA, increased only at the lowest dose. GSH content and GPx4 activity were higher than control in all treatment groups. Gene expression of *keap1* and *nrf2* transcription factors showed a dual response, down-regulation followed by an up-regulation in the lowest and the highest dose groups. Expression of *gpx4a* and *gpx4b* genes showed down-regulation, followed by up-regulation irrespective of the dose. There is little information about the effect of STC on lipid peroxidation and antioxidant defense in fish [2]; however, in vivo studies performed on rats and chickens showed an increase in CAT activity in rats [40,41] and GPx activity in chicken [42]. Our previous short-term STC exposure study with common carp showed that different doses (1 mg, 2 mg, and 4 mg STC /kg feed) have a moderate effect on lipid peroxidation parameters in the liver, and only at low dose. GSH content also increased as the lowest STC dose, while GPX4 activity decreased at a medium dose level. Expression of *keap1*, *nrf2*, *gpx4a*, *gpx4b,* and *gss* genes revealed an early down-regulation and later induction [43]. The purpose of the present study was to examine the short-term (24 h) individual or combined effects of STC and AFB1 contaminated diet on lipid peroxidation, glutathione redox system parameters, and expression of genes encoding their synthesis or metabolism in the liver of one-year-old common carps.

## 2. Results

Mortality was not observed during the 24 hour-long trial in the experimental groups. Markers of the initial phase of lipid peroxidation, conjugated dienes (CD) and trienes (CT), showed similar changes; however, the applied doses only slightly affected these parameters. The levels of CD in the AFB1 treated group were significantly higher than in the STC group 8 h after exposure. Later, 24 h after exposure, CD values of the STC treatment group were significantly lower than the control, but the CD level increased in control during the experimental period. Treatment and sampling time effects showed significant differences, but the treatment x sampling time effect was not significant. In CT, 8 h after exposure, significantly higher values were observed in the AFB1 group than STC, AFB1 + STC, and control groups. In contrast, 24 h after exposure, significantly lower values were measured as the effect of STC compared to control. However, the CT level increased in control during the experimental period. Treatment, sampling time, and treatment x sampling time effects were significantly different. Thiobarbituric acid reactive substances expressed as MDA, and the carp liver’s content did not change during the 24-h experiment. Treatment and treatment x sampling time effects were not significant, but sampling time had a significant effect (Table 1).

The antioxidant defense system parameters reduced glutathione (GSH) content, and glutathione peroxidase 4 activity (GPx4) was only slightly activated. Significantly lower GSH values were observed as an effect of STC 8 h after exposure, and there was no significant difference among the treatment groups after that. However, individual treatment and sampling time effects were significant all through the treatment, and the treatment x sampling time effect was not significant. GP×4 activity did not change significantly among the experimental groups during the 24-h trial. Treatment and treatment x sampling time effects were not significant, but the sampling time showed a significant difference (Table 2).

The gene expression of *nrf2* increased significantly as the effect of AFB1 and AFB1+STC at 8-h sampling compared to the control and those groups that were individually treated with AFB1 or STC. 16 h after mycotoxin exposure, gene expression was significantly higher than the control in the AFB1+STC group, while in the AFB1 group, significantly lower values were observed. Eight hours later, at 24th hour sampling, significantly lower values were measured as the effect of AFB1 compared to STC, AFB1+STC, and control treatments. In contrast, as an effect of AFB1+STC, significantly higher values were observed than in the other experimental groups. The overall effect of treatment, sampling time, and treatment x sampling time were significant (Table 3).

The relative expression of the *keap1* gene was significantly lower than the control as the effect of AFB1, but significantly higher as the effect of STC at 8 h. After 16 h of exposure, gene expression was significantly higher than in control as the effect of AFB1 + STC, while at the 24 h a significantly lower value was observed as an effect of AFB1, and a significant increase was measured as an effect of AFB1 + STC when compared to control. The overall effect of treatment, sampling time, and treatment x sampling time was statistically significant (Table 3).

The *gpx4a* gene expression was significantly lower than the control 8 h after exposure as an effect of STC and AFB1 + STC. Still, it decreased in all mycotoxin-treated groups 16 h after exposure compared to the control. At 24 h, the highest relative gene expression was found as an effect of STC, and the lowest as an effect of AFB1. Treatment, sampling time, and treatment x sampling time effects were significant (Table 4). 

The expression of *gpx4b* was significantly higher as the effect of AFB1 treatment during the 24-h long experiment compared to the control and STC group, while the AFB1+STC group showed significantly higher values at 16 h and 24 h as compared to STC treated group. Treatment, sampling time, and treatment x sampling time effects were significant (Table 4).

The relative expression of the *gss* gene was significantly higher as the effect of AFB1 and AFB1+STC compared to control 8 h after exposure. After 16 h of exposure, gene expression was significantly lower as the effect of AFB1 and STC compared to control. After 24 h of mycotoxin-exposure, significantly lower values were observed as an effect of STC and AFB1+STC compared to control. Treatment, sampling time, and treatment x sampling time effects were significant (Table 5). 

The *gsr* gene expression was significantly higher as the effect of AFB1 and AFB1+STC compared to control 8 h after exposure. At the 16 h, gene expression of the STC treated group showed significantly lower values than all the other experimental groups. At 24 h after exposure, the *gsr* gene expression was significantly lower in all treatment groups than in control. Treatment, sampling time, and treatment × sampling time effects were significant (Table 5).

## 3. Discussion

Our previous in vivo, short-term (24 h) studies demonstrated that individually applied doses of AFB1 and STC significantly increase the formation of oxygen free radicals in the liver of one-year-old common carp [39,43]. Oxidative stress is proposed as the effect of both AFB1 [29] and STC toxicity [28], and the oxidative reactions activate lipid peroxidation [40,41,42,43]. The results of the present study partly supported these findings because the initial phase of lipid peroxidation, the amount of CD and CT, increased as an effect of AFB1 treatment 8 h after exposure. However, lipid peroxidation did not reach the termination phase, as proven by the non-significant changes in MDA content, possibly due to the short-term period of exposure or the concentration of endogenously synthesized antioxidants (GSH and GPx4), which were readily present to inhibit the propagation of lipid peroxidation. These results suggested that the cellular defense mechanism inhibited the lipid peroxidation processes at the applied doses. However, these are contrary to our previous results with an individually applied dose of AFB1 and STC, where a significant increase of CD, CT, and MDA values was observed 16 h after exposure [39,43]. The present study results revealed that the individual effects of AFB1 and STC did not differ significantly from the AFB1+STC combination. It should be noted that sampling time has a significant effect on CD and CT levels, even in the control group. These differences are possibly caused by the rate of absorption and metabolism of nutrients from the gut during the study period because a single oral dose of feed was used after 18 h of fasting. According to the results of our previous study [39], the transit time of feed particles at 19 °C water temperature in young carp is 16 h; therefore, the gut was nearly empty at the start of the experiment.

Changes in the lipid peroxidation parameters can be explained by the changes in the glutathione redox parameters because neither GSH content nor GPx activity increased systematically during the period of AFB1, STC, or AFB1 + STC exposure. However, sampling time had a significant effect on GSH content and GPx activity, particularly 8 h after feeding, which can be explained by the absorption and metabolism of nutrients, as mentioned before. The lack of changes in the amount or activity of the glutathione redox system means that it was able to eliminate oxygen free radicals resulting from 24 h of mycotoxin exposure. These findings were in line with our previous studies when AFB1 or STC was applied individually [39,43]; therefore, exposure of AFB1 and STC together did not have an additional effect.

Looking at *nrf2*, a master regulator of the antioxidant response controlling the cytoprotective defense system [44] and its expression was upregulated as an effect of AFB1+STC during the 24-h long trial. Still, individual effects of AFB1 or STC were different because down-regulation or control levels were observed. The relative gene expression level also changed as a function of time, possibly due to metabolites generated from nutrients absorbed. The results revealed that AFB1 and STC, in combination, have a synergistic effect for the induction of *nrf2* gene expression. In an experiment with broiler chicken, AFB1 exposure also caused the down-regulation of Nrf2 at mRNA and protein levels and down-regulated the xenobiotic transformation phase II genes, such as *gst* [45]. Therefore, low *nrf2* expression as an effect of individual exposure of AFB1 and STC may have led to the down-regulation of antioxidant gene clusters such as *gpx4a*, *gpx4b*, *gss,* and *gsr* in short-term exposure. The expression of the *keap1* gene, which plays a role in the ubiquitination and degradation of Nrf2, showed dual response during the trial as an inhibition was observed at 8 and 16 h sampling, which was followed by an induction 24 h after exposure as the effect of AFB1+STC. This result shows a synergistic effect between AFB1 and STC when added together in *keap1* expression. Time-dependent changes were also found in the control group, which can be explained as the effect of metabolites of absorbed nutrients, as mentioned above. 

Among the GPx isoenzymes, glutathione peroxidase 4 (*GPX4*) gene expression has primary importance in the antioxidant defense of the avian and fish species [46,47,48], while in mammals GPX1 plays the major role [49]. The expression of *gpx4a* and *gpx4b* showed opposite changes during the trial. In the case of *gpx4a*, a continuous down-regulation, while in the case of *gpx4b*, a continuous upregulation was observed as an effect of AFB1. STC, but AFB1 + STC have the opposite effect in some cases. The results revealed that the effect of AFB1 + STC seems to be antagonistic with the individual effect of AFB1 and synergistic with the individual effect of STC. The expression of *gss* and *gsr* genes also showed dual response as induction was observed 8 h after exposure as an effect of AFB1 and AFB1+STC in the case of both *gss* and *gsr*. A downregulation followed these inductions in the case of all treatment groups. The alterations in the expression of *gss* and *gsr* may explain the changes in the GSH level. The exposure with AFB1 + STC combination modified the individual effect of AFB1 and STC only moderately, but in STC, a synergistic effect can be hypothesized. Relative expression of GPx isoenzymes also showed changes, even in the control group, as a function of time. These changes can be explained by the support of amino acids and selenium from the diet, as mentioned before.

## 4. Conclusions

In conclusion, the results revealed that the individual effects of AFB1 and STC on biochemical parameters are similar to combined treatment. On the contrary, at the gene expression level, some synergistic (*keap1*, *nrf2, gpx4a, gss, gsr*) or antagonistic (*gpx4a*, *gpx4b*) effects can be hypothesized as AFB1 and STC have given in combination. These results will be important for evaluating multi-mycotoxin exposure in common carp.

## 5. Materials and Methods 

### 5.1. Production of Mycotoxins and Analyses

AFB1 production was performed by a toxicogenic *Aspergillus flavus* strain (SZMC 20750) isolated by Dobolyi et al. [50] on corn substrate (measured concentration of fungal culture was 4694 mg AFB1 and 28.1 mg AFB2/kg dry matter), while STC was bought from Romer Labs (Tulln, Austria) and was of high purity (99.0 ± 1.0%). 

### 5.2. Experimental Design, Sample Preparations, Biochemical Determinations

In the present trial, 78 one-year-old common carp (*Cyprinus carpio L.*) juveniles (body weight: 49.07 ± 8.85 g) purchased from a commercial fish farm (Balaton Fish Management Non-Profit Ltd., Buzsák-Ciframalon, Hungary) were used. A week-long acclimatization period was applied before randomly dividing the fish into four treatment groups (namely: control, aflatoxin, sterigmatocystin, AFB1 + STC) into four aquariums (150 L each). The aquariums were used in a semi-static system filled with dechlorinated tap water that was continuously aerated. During the experiment, the water temperature was 19 ± 1 °C, which defines the moderate basal metabolic rate for common carp. The light regimen was set to 12 h light: 12 h dark. The following dosage groups were used to investigate the effect of both individual and mycotoxin mixture: control; AFB1 (100 µg AFB1/kg feed); STC (1000 µg STC/kg feed); AFB1+STC (100 µg AFB1 and 1000 µg STC/kg feed). The dose range was selected based on our previous short-term studies with individual AFB1 [39] or STC [43] exposure in carp. In the case of AFB1, an appropriate amount of mycotoxin-containing fungal culture was mixed with ground growth feed for carp (GARANT Aqua Classic™, Garant-Tiernährung, Pöchlarn). The nutrient content of the diet was 30% crude protein, 7% crude fat, 5% crude fiber, 7.5% crude ash, and 50.5% nitrogen free extract (on dry matter basis). The mycotoxin concentration of the control diet was <1.0 μg AFB1 and <1.0 μg AFB2/kg. In the purified sterigmatocystin-contaminated diet, sterigmatocystin was dissolved in absolute ethanol and mixed with the growth feed for carp, making a stock mixture.

The measured mycotoxin concentration of the experimentally contaminated diet was 93.5 μg AFB1 and 0.6 μg AFB2/kg. The experimental diets were diluted to 5 mL water immediately before use. The mycotoxin exposure of the fish in the experimental groups was calculated according to the individual body weight and the amount of experimentally contaminated diet that contained the intended mycotoxin content in the particular group. The calculated mycotoxin intake of the animals was 0.95 μg AFB1/kg body weight in the aflatoxin treated group, 10.06 μg STC/ kg body weight in the sterigmatocystin treated group, and 0.95 μg AFB1/kg body weight + 10.27 μg STC/ kg body weight in the aflatoxin + sterigmatocystin treated group, respectively. 

Six fish served as absolute control (0 h) at the beginning of the experiment. Control and experimentally contaminated feed were given by gavage to the gut once. Fish were over-anaesthetized with clove oil and decapitated. Liver samples were taken from 6 carps of each group 8, 16, and 24 h after exposure, immediately frozen in liquid nitrogen, and stored at −80 °C until analysis preventing RNA degradation. Markers of the lipid peroxidation (conjugated dienes, trienes, and thiobarbituric acid reactive substances), and reduced glutathione (GSH) concentration, and glutathione peroxidase 4 (GPx4) activity, were measured without modifications as described previously [41].

### 5.3. RNA Isolation, Reverse Transcription, and Qpcr

Total RNA was extracted from the liver of six fish from each group. The RNA sample preparation, qPCR procedure, and relative RNA abundance qualification were conducted without modifications as previously described [39]. Primers (Table 6) for the glutathione peroxidase 4a, 4b (*gpx4a, gpx4b*), glutathione synthetase (*gss*), glutathione reductase (*gsr*), nuclear factor-erythroid 2 p45-related factor 2 (*nrf2*), kelch-like ECH-Associated protein 1 (*keap1*) and reference gene *β-actin* were chosen based on the literature [51,52,53].

### 5.4. Statistical Analysis

All data are presented as mean ± standard deviation (SD). Firstly, the data were tested by the Shapiro-Wilk normality test, and to confirm the homogeneity of variance both Bartlett and Browne-Forsythe tests were performed.

Two-factor ANOVA was used, followed by the Tukey method for multiple comparisons to assess the effects of the applied doses and exposure time. The significance level was set at *P* < 0.05. Statistical analyses were conducted with the GraphPad Prism 7.0 software (GraphPad Software, San Diego, CA, USA).

### 5.5. Ethical Issues

The experiment was carried out according to the Hungarian Animal Protection Act, in compliance with the relevant EU rules. The experimental protocol was authorized by the Department of Food Chain Safety, Land Register, Plant and Soil Protection and Forestry of the Pest County Government Office (Budapest, Hungary) with a permission number PE/EA/1964-7/2017 (approval date: 7 December 2017; the permit is valid for five years).

## Figures and Tables

**Table 1 toxins-13-00109-t001:** Effect of aflatoxin B1 (AFB1), sterigmatocystin (STC), and AFB1+STC treatment on parameters of lipid peroxidation in carp liver homogenates (mean ± S.D.; n = 6).

Conjugated dienes (OD 232 nm)					
	0 h	8 h	16 h	24 h	*p*-value
Control	0.15 ± 0.11 ^A^	0.28 ± 0.09 ^abAB^	0.38 ± 0.13 ^B^	0.41 ± 0.08 ^bB^	T: 0.0202 H: <0.0001 T × H: 0.1647
AFB1		0.41 ± 0.17 ^bB^	0.42 ± 0.08 ^B^	0.28 ± 0.11 ^abAB^	
STC		0.20 ± 0.06 ^aAB^	0.35 ± 0.13 ^B^	0.20 ± 0.06 ^aAB^	
AFB1+STC		0.26 ± 0.03 ^abAB^	0.36 ± 0.11 ^B^	0.24 ± 0.06 ^abAB^	
**Conjugated trienes (OD 268 nm)**					
	0 h	8 h	16 h	24 h	
Control	0.07 ± 0.05 ^A^	0.14 ± 0.04 ^aAB^	0.17 ± 0.05 ^B^	0.21 ± 0.04 ^bB^	T: 0.0145 H: <0.0001 T × H: 0.0181
AFB1		0.23 ± 0.07 ^bC^	0.20 ± 0.04 ^BC^	0.14 ± 0.06 ^abAB^	
STC		0.10 ± 0.03 ^aAB^	0.18 ± 0.06 ^B^	0.10 ± 0.03 ^aAB^	
AFB1+STC		0.13 ± 0.02 ^aAB^	0.18 ± 0.06 ^B^	0.13 ± 0.03 ^abAB^	
**Thiobarbituric acid reactive substances (malondialdehyde μmol/g wet weight)**					
	0 h	8 h	16 h	24 h	
Control	16.87±7.91 ^A^	24.22 ± 12.51	32.12 ± 21.00	17.33 ± 5.30	T: 0.5172 H: 0.0001 T × H: 0.9393
AFB1		29.86 ± 10.88 ^AB^	36.76 ± 21.61 ^B^	14.98 ± 7.33 ^A^	
STC		15.67 ± 7.27	30.35 ± 18.41	14.54 ± 5.59	
AFB1+STC		23.87 ± 6.53	27.27 ± 11.91	15.53 ± 4.92	

^a,b^ Different superscripts within columns mean significant difference (*p* < 0.05) between treatment groups. ^A,B,C^ Different capital letters within a row mean significant difference between sampling times (*p* < 0.05). T = treatment effect; H = time effect; T × H = treatment × time effect.

**Table 2 toxins-13-00109-t002:** Effect of aflatoxin B1 (AFB1), sterigmatocystin (STC), and AFB1+STC treatment on the amount/activity of glutathione redox system of 10,000 g supernatant fraction of carp liver homogenates (mean ± S.D.; n = 6).

Reduced glutathione (μmol/g protein content)					
	0 h	8 h	16 h	24 h	*p*-value
Control	4.31 ± 2.19 ^A^	10.02 ± 2.39 ^abB^	7.90 ± 1.90 ^B^	7.80 ± 1.51 ^B^	T: 0.0352 H: <0.0001 T × H: 0.4216
AFB1		9.21 ± 2.50 ^abB^	7.28 ± 1.62 ^AB^	6.81 ± 2.11 ^AB^	
STC		5.82 ± 0.77 ^aA^	6.64 ± 1.85 ^A^	6.19 ± 1.00 ^A^	
AFB1+STC		7.36 ± 0.96 ^bAB^	7.81 ± 2.01 ^B^	7.24 ± 1.65 ^AB^	
**Glutathione peroxidase** **(U/g protein content)**					
	0 h	8 h	16 h	24 h	
Control	4.28 ± 2.86 ^A^	8.49 ± 2.65 ^B^	9.87 ± 2.69 ^B^	9.86 ± 2.03 ^B^	T:0.2365 H: <0.0001 T × H: 0.3600
AFB1		10.91 ± 2.41 ^B^	8.70 ± 2.13 ^B^	8.52 ± 2.60 ^B^	
STC		7.28 ± 1.98 ^AB^	9.67 ± 2.53 ^B^	7.64 ± 1.87 ^AB^	
AFB1+STC		8.80 ± 0.49 ^B^	11.88 ± 1.46 ^B^	10.13 ± 2.39 ^B^	

^a,b^ Different superscripts within columns mean significant difference (*p* < 0.05) between treatment groups. ^A,B^ Different capital letters within a row mean significant difference between sampling times (*p* < 0.05). T = treatment effect; H = time effect; T × H=treatment × time effect.

**Table 3 toxins-13-00109-t003:** Effect of aflatoxin B1 (AFB1) sterigmatocystin (STC), and AFB1+-STC treatment on the relative expression of *nrf2* and *keap1* genes in the liver of common carp (mean ± S.D.; n = 6 in a pool, equal amounts of cDNA per individual).

Nuclear Factor-Erythroid 2 p45-Related Factor 2 (*nrf2)*					
	0 h	8 h	16 h	24 h	*p*-value
Control	1.00 ± 0.03 ^A^	0.94 ± 0.07 ^aA^	1.84 ± 0.18 ^bC^	1.57 ± 0.12 ^cB^	T: <0.0001 H: <0.0001 T × H: <0.0001
AFB1		1.23 ± 0.19 ^bA^	1.16 ± 0.18 ^aA^	0.37 ± 0.02 ^aB^	
STC		0.91 ± 0.16 ^aA^	1.72 ± 0.30 ^bB^	0.91 ± 0.10 ^bA^	
AFB1+STC		1.88 ± 0.17 ^cB^	2.65 ± 0.23 ^cD^	2.28 ± 0.32 ^dC^	
**Kelch-like ECH-Associated Protein 1 (*keap1)***					
	0 h	8 h	16 h	24 h	
Control	1.04 ± 0.33 ^A^	3.95 ± 0.17 ^bC^	2.40 ± 0.09 ^aB^	1.59 ± 0.19 ^bA^	T: <0.0001 H: <0.0001 T × H: <0.0001
AFB1		2.81 ± 0.25 ^aB^	2.58 ± 0.22 ^aB^	0.69 ± 0.1d1 ^aA^	
STC		5.07 ± 0.35 ^cD^	2.34 ± 0.24 ^aC^	1.62 ± 0.19 ^bB^	
AFB1+STC		3.61 ± 0.66 ^bB^	3.65 ± 0.36 ^bB^	4.54 ± 0.84 ^cC^	

^a,b,c,d^ Different superscripts within columns mean significant difference (*p* < 0.05) between treatment groups. ^A,B,C,D^ Different capital letters within a row mean significant difference between sampling times (*p* < 0.05). T = treatment effect; H= time effect; T × H = treatment × time effect.

**Table 4 toxins-13-00109-t004:** Effect of aflatoxin B1 (AFB1), sterigmatocystin (STC) and AFB1+STC treatment on the relative expression of *gpx4a* and *gpx4b* genes in the liver of common carp (mean ± S.D.; n = 6 in a pool, equal amounts of cDNA per individual).

Glutathione peroxidase 4a (*gpx4a)*					
	0 h	8 h	16 h	24 h	*p*-value
Control	1.00 ± 0.07 ^A^	1.51 ± 0.06 ^cB^	1.46 ± 0.02 ^cBC^	1.34 ± 0.05 ^cC^	T: <0.0001 H: <0.0001 T × H:<0.0001
AFB1		1.49 ± 0.18 ^cD^	0.72 ± 0.07 ^aC^	0.36 ± 0.04 ^aB^	
STC		0.91 ± 0.08 ^aA^	1.17 ± 0.07 ^bB^	1.97 ± 0.16 ^dC^	
AFB1+STC		1.17 ± 0.07 ^bB^	1.16 ± 0.09 ^bB^	0.68 ± 0.02 ^bC^	
**Glutathione peroxidase 4b (*gpx4b)***					
	0 h	8 h	16 h	24 h	
Control	1.00 ± 0.10 ^AB^	0.80 ± 0.14 ^aA^	1.21 ± 0.09 ^bB^	0.95 ± 0.06 ^aA^	T: <0.0001 H: <0.0001 T × H: <0.0001
AFB1		1.29 ± 0.16 ^cC^	3.95 ± 0.31 ^cE^	2.90 ± 0.31 ^cD^	
STC		0.87 ± 0.09 ^abAC^	0.79 ± 0.13 ^aAC^	0.74 ± 0.05 ^aC^	
AFB1+STC		1.05 ± 0.08 ^bA^	1.29 ± 0.17 ^bC^	1.22 ± 0.18 ^bAC^	

^a,b,c,d^ Different superscripts within columns mean significant difference (*p* < 0.05) between treatment groups. ^A,B,C,D,E^ Different capital letters within a row mean significant difference between sampling times (*p* < 0.05). T = treatment effect; H = time effect; T × H = treatment × time effect.

**Table 5 toxins-13-00109-t005:** Effect of aflatoxin B1 (AFB1), sterigmatocystin (STC) and AFB1+STC treatment on the relative expression of *gss* and *gsr* genes in the liver of common carp (mean ± S.D.; n = 6 in a pool, equal amounts of cDNA per individual).

Glutathione synthetase (*gss)*					
	0 h	8 h	16 h	24 h	*p*-value
Control	1.01 ± 0.12 ^AB^	0.84 ± 0.15 ^aAB^	1.09 ± 0.15 ^bAB^	1.03 ± 0.15 ^cAB^	T: <0.0001 H: <0.0001 T × H: <0.0001
AFB1		1.25 ± 0.33 ^bA^	0.57 ± 0.15 ^aC^	0.89 ± 0.07 ^bcB^	
STC		0.60 ± 0.19 ^aC^	0.67 ± 0.20 ^aC^	0.29 ± 0.08 ^aD^	
AFB1+STC		3.35 ± 0.23 ^cD^	1.18 ± 0.22 ^bA^	0.68 ± 0.15 ^bC^	
**Glutathione reductase (*gsr)***					
	0 h	8 h	16 h	24 h	
Control	1.01 ± 0.13 ^A^	0.89 ± 0.17 ^aA^	1.49 ± 0.38 ^bB^	1.17 ± 0.15 ^bA^	T: <0.0001 H: <0.0001 T × H: <0.0001
AFB1		2.21 ± 0.18 ^cD^	1.60 ± 0.18 ^bC^	0.73 ± 0.16 ^aB^	
STC		0.86 ± 0.14 ^aA^	0.77 ± 0.22 ^aAB^	0.58 ± 0.13 ^aB^	
AFB1+STC		1.44 ± 0.22 ^bB^	1.46 ± 0.18 ^bB^	0.79 ± 0.15 ^aA^	

^a,b,c^ Different superscripts within columns mean significant difference (*p* < 0.05) between treatment groups. ^A,B,C,D^ Different capital letters within a row mean significant difference between sampling times (*p* < 0.05). T = treatment effect; H= time effect; T × H = treatment × time effect.

**Table 6 toxins-13-00109-t006:** Primers of endogenous control (*β-actin*) and target (*gpx4a, gpx4b, gss, gsr, nrf2* and *keap1*) genes.

Gene	Primers		Accession Nr.
	Forward (5′–3′)	Reverse (5′–3′)	
*β-actin*	GCAAGAGAGGTATCCTGACC	CCCTCGTAGATGGGCACAGT	XM_019103102.1
*gpx4a*	GGAACCAGGAACAAATTCCC	AGATCCTTCTCCACCACGCTTG	FJ656211.1
*gpx4b*	CTACAAGGCAGAGTTTGACCTC	CTTGGATCGTCCATTGGTCC	FJ656212.1
*gss*	ACCATGACATACCGCTGACAT	TGTTCCCCATAGATCAGTAGAGGAT	XM_019114684.1
*gsr*	ACTCGTGCAGGTGTCTATGC	TTTGGAGTCTGCTTTGCCCT	HQ174244.1
*nrf2*	TTCCCGCTGGTTTACCTTAC	CGTTTCTTCTGCTTGTCTTT	JX462955
*keap1*	GCTCTTCGGAAACCCCT	GCCCCAAGCCCACTACA	JX470752

## Data Availability

The raw data supporting the conclusions of this manuscript will be made available by the authors, without undue reservation, to any qualified researcher.

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
