# Peer review of "Individual and Combined Effects of Aflatoxin B1 and Sterigmatocystin on Lipid Peroxidation and Glutathione Redox System of Common Carp Liver"

_toxins, 2021, doi:10.3390/toxins13020109_

Round 1

Reviewer 1 Report

The study is very interesting. I recommend just very few grammar corrections

  1. row 107- split the word parameters in para-meters
  2. rows 138 and 149 split the word significant in signi-ficant or signifi-cant

Author Response

Reply to Reviewer 1.

Thank you for the positive opinion about our manuscript.

The Authors made the proposed editorial changes by track changes.

row 107- split the word parameters in para-meters -  corrected

rows 138 and 149 split the word significant in signi-ficant or signifi-cant -  corrected

Reviewer 2 Report

Individual and combined effects of aflatoxin B1 and sterigmatocystin on lipid peroxidation and glutathione redox system of common carp liver

Toxins January 2021

This paper aims to use oxidative stress measurements in the liver of carp exposed to the mycotoxins aflatoxin B1 and sterigmatocystin. A time course analysis was conducted of the expression of oxidative stress markers thiobarbiuric acid reactive substances (TBARS), conjugated dienes and conjugated trienes as well as reduced glutathione and glutathione peroxidase. Gene expression of nrf-2, keap1, glutathione peroxidases gpx4a and gpx4b, glutathione synthase and glutathione peroxidase. Whilst reasonably well written some grammar requires extensive editing.

Major issues.

  1. It is unclear how much if any AFB1 the fish were subjected to. The details in the methodology are minimal and need to be improved, for all methods. Further details regarding the composition of the fish feed is important and the concentrations of the aflatoxin that were produced by the Aspergillus species including other toxins other than AFB1.
  2. The control data for all compounds, enzymes and most genes tested vary significantly between time points. This has not been addressed in the text. A small variation between samples and time points is understandable and expected in biological samples, as well as variations in genes/enzymes due to circadian rhythms or growth etc. However, there is a lot of variability in the control data and most of the results for the control samples would indicate that there is significant oxidative stress in the fish after the initial samples were taken. The variation in control data makes it extremely difficult to justify the conclusions that are drawn from the data and may also help explain the contrary results obtained with this analysis as compared to previous analyses.
  3. The presentation and analysis of time course data is a difficult task, comparison of each time point to control is considered to be an appropriate method for large data sets. The data presented is not a large dataset and could also be plotted on x-axis time scale. To be able to better understand the time course data, time 0 points from the treated samples need to be taken into account. Overall, it is very difficult to interpret the data as it is presented and conflicting results to other studies appears to be due to poorly collected data.
  4. The materials and methods section needs complete details of all experimental methods, at least in brief and/or with modifications from other established methods.

Some minor points

Requires consistency with editing such as

Line 44 – In vivo and in vitro should be in italics

Use of hours 12 h instead of 12h. e.g. Line 269

Editing such as

Line 27 Due to increased fish production in aquaculture

Lines 28 to 29 could have better sentence structure

Line 51 Metabolised by cytochrome P450

etc

Line 32 remove “which is a group of more than 20 compounds”. Does this include metabolites as well as products produced from

Round 2

Reviewer 2 Report

The authors have addressed the major concerns regarding the large variation in controls and missing details in the methodology. The authors have also edited the paper for grammar and addressed other minor concerns.

There are no other concerns for this paper to go to publication.